# Inclusion of photoautotrophic cultivated diatom biomass in salmon feed can deter lice

**Hans Chr. Eilertsen**[ID][1]*, **Edel Elvevoll**[1], **Ingeborg Hulda Giæver**[1], **Jon Brage Svenning**[1], **Lars Dalheim**[1], **Ragnhild Aven Svalheim**[2], **Birthe Vang**[2], **Sten Siikavuopio**[2], **Ragnhild Dragøy**[2], **Richard A. Ingebrigtsen**[ID][1], **Espen Hansen**[1], **Anette Hustad**[2], **Karl-Erik Eilertsen**[ID][1]

**1** Norwegian College of Fishery Science, UiT The Arctic University of Norway, Tromsø, Norway, **2** NOFIMA Norwegian Institute of Food, Fisheries and Aquaculture Research, Tromsø, Norway

* hei000@uit.no

**Data Availability Statement:** All relevant data are within the manuscript and its Supporting Information files.

## Abstract

The aim of this study was to evaluate the potential of diatom (microalgae) biomass as a lice-reducing ingredient in salmon feed. The original hypothesis was based on the fact that poly-unsaturated aldehydes (PUAs), e.g. 2-trans, 4-trans decadenial (A3) produced by diatoms can function as grazing deterrents and harm copepod development. Salmon lice (*Lepeophtheirus salmonis*) is a copepod, and we intended to test if inclusion of diatom biomass in the feed could reduce the infestation of lice on salmon. We performed experiments where salmon kept in tanks were offered four different diets, i.e. basic feed with diatoms, fish oil, *Calanus* sp. oil or rapeseed oil added. After a feeding period of 67 days a statistically representative group of fishes, tagged with diet group origin, were pooled in a 4000L tank and exposed to salmon lice copepodites whereafter lice infestation was enumerated. Salmon from all four diet groups had good growth with SGR values from 1.29 to 1.44% day$^{-1}$ (increase from ca. 130 g to 350 g). At the termination of the experiment the number of lice on salmon offered diatom feed were statistically significantly lower than on salmon fed the other diets. Mean lice infestation values increased from diatom feed through *Calanus* and fish oil to standard feed with terrestrial plant ingredients. Analysis of the chemical composition of the different diets (fatty acids, amino acids) failed to explain the differences in lice infestation. The only notable result was that diatom and *Calanus* feed contained more FFA (free fatty acids) than feed with fish oil and the control feed. None of the potential deleterious targeted polyunsaturated aldehydes could be detected in skin samples of the salmon. What was exclusive for salmon that experienced reduced lice was diatom inclusion in the feed. This therefore still indicates the presence of some lice deterring ingredient, either in the feed, or an ingredient can have triggered production of an deterrent in the fish. An obvious follow up of this will be to perform experiments with different degrees of diatom inclusion in the feeds, i.e. dose response experiments combined with targeted PUA analyses, as well as to perform large scale experiments under natural conditions in aquaculture pens.

**Funding:** This study was financed by The Norwegian Seafood Research Fund (FHF).

**Competing interests:** The authors have declared that no competing interests exist.

## Introduction

Salmonid aquaculture industry in Norway experiences significant challenges relating to environmental impacts, fish health, escape from pens and salmon lice (*Lepeophtheirus salmonis*) [1–3]. These culprits may limit the industry's ability to grow. Salmon lice can cause disease and reduce growth, and at high concentrations it can even cause death [4]. In order to protect wild salmonid stocks and farmed fish, it will be necessary to reduce lice infestations. Therefore the Norwegian government has issued strict regulations, i.e. a maximum prevalence of 0.2 female lice per fish during spring is allowed. This has triggered development and use of a plethora of delousing methods, i.e. chemical and mechanical as well as cleaner fish. These delousing methods have in fact increased an already too high mortality in Norwegian salmon aquaculture [5]. The use of cleaner fish often has, if any, very short—lived effects [6]. The annual total cost of delousing in Norway now has passed 6 billion NOK, not including losses related to reduced growth and exaggerated mortality [7].

The inclusion of marine oil in Norwegian fish feed has the recent years decreased since marine ingredients have gradually been replaced with alternatives, primarily of terrestrial agricultural origin [8, 9]. This has led to reduced n-3/n-6 long chain polyunsaturated fatty acid (LC-PUFA) ratios in farmed salmon. Consumption of fish or seafood is recognized to reduce the risk of cardiovascular diseases in both animals and humans. This is largely based on epidemiological analysis supported by preclinical and clinical studies on effects of the long chain omega-3 (n-3) polyunsaturated fatty acids EPA (eicosapentaenoic acid; 20:5n-3) and DHA (docosahexaenoic acid; 22:6n-3). It has been documented that altered fatty acid composition in salmon feed can have negative effect on early development of tissue and organs, resistance to infections and fish health [10, 11]. The extensive use of terrestrial ingredients has also decreased the content of omega-3 LC-PUFA in Atlantic salmon, leading to a compromised nutritional value of the final product, and more than double fish portion sizes are now required to satisfy recommended EPA + DHA intake levels [1, 8]. Research and industrial development should thus give priority to re-introduce marine LC-PUFA in salmon feed as well as to reduce mortality related to delousing, not only for economic reasons, but also to improve fish welfare.

The increasing world population also needs more food [12]. As land-based resources are scarce, the contribution of sustainable food and feed from ocean resources is essential. From 1950 to 2010, 27% of global marine fisheries landings were destined for other uses (farmed fish, chicken and pigs) than direct human consumption. Also, fish catches destined for uses other than human consumption are largely food-grade or prime food-grade, strengthening the conclusion that future fish feed must have origins other than fisheries [12]. Photoautotrophic marine diatoms are here good candidates as they grow fast (>100% increase in biomass/day) and contain high levels of EPA and DHA (e.g. 9.2–32% of the total PUFA) plus substantial amounts of amino acids, vitamins and other functional substances [13, 14]. Microalgae production also has clear advantages in terms of ecological footprint (energy, $CO_2$) relative to conventional fish feed ingredients [15].

The use of feed with microalgae to deter salmon lice is based on a hypothesis generated many years ago. We originally observed that both intake of feed and lice infestation on salmonids decreased at the same time as the phytoplankton (microalgae) spring bloom progressed. We assumed this to be caused by a phytoplankton toxin [16, 17]. It was later confirmed that the culprit was a polyunsaturated aldehyde (oxylipin) produced by the haptophycean microalgae *Phaeocystis pouchetii* [18]. Oxylipins are lipids that are enzymatically produced from membrane-bound PUFAs, especially LC-PUFAs (EPA and DHA). The spring bloom in northern and temperate areas is made up by diatoms and *P. pouchetii* [19], and production of oxylipins

as copepod grazing deterrents in marine ecosystems has been shown for several diatom species [20]. The oxylipin 2-trans, 4-trans decadenial (A3) has later been documented to reduce lice infestation after injection into salmon [21]. Since salmon lice is a copepod, we hypothesized that inclusion of diatoms in the fish feed could reduce lice infestation. To test this we offered four diets of different origins to salmon, including one with diatoms, and monitored lice infestation.

## Materials and methods

### Cultivation of diatom biomass

The applied strain of the diatom *Porosira glacialis* was isolated from samples taken during a research cruise to the Arctic in 2014 and maintained in Guillard f/10 medium, as part of the culture collection at NFH at UiT—The arctic university of Norway. Cultivation temperature was ca. 4 $^{\circ}$C and photoperiod 14:10 (day:night) at scalar light intensity 20–30 µmol quanta m$^{-2}$s$^{-1}$. The mass cultivation took place in a 300 000 L vertical column photo-bioreactor integrated in the production line at the ferrosilicon producer Finnfjord AS at Finnsnes in Northern Norway. Illumination (continuous) was natural light and daylight (white) LEDs with a mean light intensity in the reactor of ca. 25 µmol m$^{-2}$s$^{-1}$, i.e. 6 W m$^{-2}$. The biomass was produced during several three and four day periods in September and October 2018, and the culture was supplied with $CO_2$ by bubbling with factory smoke, keeping pH between 7.5 and 8.2. Cultivation was semi-continuous, and 50 000–70 000 L of culture was harvested daily. The harvested culture-seawater was then replaced by 1 µm filtrated and UV-sterilized seawater from the adjacent fjord area. Inorganic macro- (N, P, Si) and micro-nutrients were Kristalon fertilizer (Yara Norway) and (Si) sodium metasilicate pentahydrate. Analysis of nutrients were performed every second day to avoid nutrient starvation. Cell concentration was monitored daily, and the culture was maintained at growth rates between 0.25 and 0.38 doublings day$^{-1}$. Biomass was harvested in two steps with a drum plankton net system and a bowl separator that concentrated the biomass to a thick paste. The harvested biomass was stored at −20˚C prior to preparation of the different salmon feed types.

### Experimental fish and lice

The experiments were performed at the Aquaculture research station in Tromsø, Norway between 06.06.2019 and 27.08.19. Atlantic salmon (*Salmo salar L.*) originating from the Aqua Gen (Qtl-innOva SHIELD) strain, were prior to the experiment maintained at the production facility in 2000L tanks in a freshwater flow-through system. A total number of 450 Atlantic salmon, ca. 130–140 g juveniles were individually tagged using pit-tag (Biomark). 10 fish were sampled for skin analysis. After ca. 17 days recovery period after tagging, the fish were randomly distributed into 12 separate tanks. Each of four feed treatments (Table 1), i.e. A-Algae, B-Fish oil, C-*Calanus* and D-Control hence had 3 replicates of 37 fish in 500 L seawater flow-through tanks. At transfer from production to experimental unit the fish were anaesthetized using Benzoak vet. (ACD Pharmaceuticals AS) before registration of tag ID, weight and fork length measurements. At start of the experiment mean fish weight was 134.9 ± 15.4 g (T0). Experimental tanks were supplied with brackish water (15 ‰, and 9.6–11˚C), and fish were given standard commercial feed (Nutra Olympic, Skretting) diet for five days before the experimental diets were introduced, with the fish kept at normal salinity (32.5 ppt) seawater and 10˚C.

After 34 days of feeding (T1) with the experimental diets, fish were anaesthetized, measured and weighed. Skin samples from 3 fish from each tank were taken and kept in 3.6 ml cryotubes, placed in dry ice. Fish that had not gained weight between T0 and T1 were not selected for

**Table 1. Ingredient composition in the four feed types, % of dry weight.**

| Diet | A-Algae | B-Fish oil | C-*Calanus* | D-Control |
|---|---|---|---|---|
| Fish meal | 25.00 | 25.00 | 25.00 | 25.00 |
| SPC | 19.50 | 19.50 | 19.50 | 19.50 |
| Wheat gluten | 17.00 | 17.00 | 17.00 | 17.00 |
| Wheat | 10.40 | 10.40 | 10.40 | 10.40 |
| Fish oil | 19.30 | 21.30 | 19.17 | 9.00 |
| Rapeseed oil | | | | 12.30 |
| *Calanus* oil | | | 2.13 | |
| Diatoms | 2.00 | | | |
| Soya lecithin | 0.50 | 0.50 | 0.50 | 0.50 |
| Choline chloride | 0.50 | 0.50 | 0.50 | 0.50 |
| Vitaminpremix | 2.00 | 2.00 | 2.00 | 2.00 |
| Monosodiumphosphate (26% P) | 2.50 | 2.50 | 2.50 | 2.50 |
| Carop. Pink (10% Astaxanthin) | 0.05 | 0.05 | 0.05 | 0.05 |
| L-Lysine | 0.60 | 0.60 | 0.60 | 0.60 |
| DL-Methionin | 0.10 | 0.10 | 0.10 | 0.10 |
| Mineral premix | 0.50 | 0.50 | 0.50 | 0,50 |
| Vitamin C (35%) | 0.05 | 0.05 | 0.05 | 0.05 |
| Sum (%) | 100.00 | 100.00 | 100.00 | 100.00 |

skin sampling. When all fish had been registered, the skin samples were flushed with nitrogen and stored at -80˚C for later analysis. After another 32 days of feeding, new registrations of growth as well as skin samples were taken on day 67 (T2) after experiment start (T0) following same procedure as for T1. In addition, 6 fish pr. tank were removed from the experiment to adjust for biomass for the salmon lice exposure study. The day after T2 sampling, a total of 297 fishes (mean weight 340.2 ± 49.2 g) were relocated to a fish health laboratory and placed in a 4000 L tank, all groups mixed. The water was oxygenated to obtain acceptable oxygen levels in the tank, (flow 120 l min⁻¹, oxygen levels 87.1%, temperature 9.9˚C). During the period at the fish health laboratory the fish were fed standard feed, i.e. Skretting Nutra Olympic 3 mm.

Infection of the salmon was performed using a challenge model developed with the Aquaculture Research Station in Tromsø, approved by The Norwegian Food Safety Authority (ref. ID 16418). This is a bath challenge model, where the fish is challenged with a known number of lice copepodids in stagnant seawater. The research station routinely produces infective copepodids by performing controlled infections, harvesting mature eggs and incubating these until they reach the copepodid stage. The dose of infection was calculated on the basis of an expected average estimate of 10 lice per fish. The experiments were performed in the fish health laboratory at the research station, where factors such as water temperature are closely monitored and controlled. The water outlet from the laboratory was disinfected to prevent distribution of live copepodids to the environment. For more specific description of the model see [22].

The experiment was ended on day 82 (T3) after start (T0) of the experiment, and 15 days post salmon lice exposure. Fish were anaesthetized (Benzoak vet), and lice on the whole-body including gills and mouth were counted manually. After counting lice, the tag ID was registered, and new weight and fork length measurements were performed. Skin samples for analysis from 9 fish pr. diet group, frozen in cryotubes in dry ice, were flushed with nitrogen and stored in a -80˚C freezer.

Specific growth rate (SGR, % day$^{-1}$) was calculated from the equation:

$$SGR = \frac{100(lnW_1 - lnW_0)}{t} \tag{1}$$

where $W_0$ and $W_1$ are mean initial and final body weights and $t$ is number of experimental days.

## Experimental diets

The four types of feed applied were prepared with the same basic composition with respect to fish meal, SPC (soy protein concentrate), wheat gluten and wheat, vitamins and minerals (Table 1). Total oil inclusion in all diets was 21.3%, i.e. fish oil and diatom biomass in diet A, fish oil solely in B, fish oil and *Calanus finmarchicus* oil in C, and fish oil and rapeseed oil in the terrestrial plant based commercial standard diet D.

## Total lipid, lipid class and fatty acids in feed and salmon

After 67 days (T3) on the experimental diets, 9 fish were randomly selected from each group (3 from each cage), gutted and immediately filleted and skinned, kept in plastic bags on ice for 2–4 hours and stored at −70˚C until laboratory analysis within 3 months.

Extraction of lipid followed Jensen et al. [23] adapted from Folchs method [24], using 2 ml of dichloromethane:methanol (2:1 v/v) as the extractant per 100 mg of pellet or fish flesh [25]. The pellets/flesh were crushed using a mortar and pestle prior to extraction, and the biomass was extracted twice to maximize yield. The organic extracts were evaporated under nitrogen and the total lipid content was determined gravimetrically. Fatty acid methylation was performed using a method adapted from Stoffel et al. [26], using sulfuric acid as the catalyst. Briefly, 100 μl of lipid extract (10 mg ml$^{-1}$ dissolved in dichloromethane) was transferred to a 15 ml glass tube and added 800 μl of dichloromethane, 100 μl of internal standard (C17:0. 0.1 mg ml$^{-1}$) and 2 ml 10% $H_2SO_4$ in MeOH. The samples were then heated and kept at 100˚C for 1 hour, cooled, and 3 ml hexane and 3 ml 5% NaCl in $H_2O$ was added. The resulting organic phase was transferred to 4 ml glass tubes. Following evaporation, the samples were resuspended in 100 μl of hexane and transferred to GC test tubes prior to analysis. For a detailed description of the extraction and methylation procedure see [27].

Fatty acid methyl esters (FAMEs) were analyzed on a GC-FID (Agilent Technologies) coupled to a Select FAME column (length 50 m, ID 0.25 mm and FT 0.25 μm, Agilent J&W Columns), using helium as the carrier gas (1.6 ml min$^{-1}$). The fatty acids were quantified based on the peak area of the chromatograms divided by the area of the internal standard and converted to absolute amounts using the slopes calculated from standard curves (triplicates of 7.8125–2000 μg ml$^{-1}$ of GLC 502 Free Acids, Nu-Check-Prep, Elysian, MN, USA).

The lipid class composition was analyzed by normal phase HPLC, using a Water e2795 separations module coupled to a Supelcosil™ LC-SI 5 mm (25 cm x 4.6 mm) column (Supelco HPLC products, Bellefonte, PA, USA) set to a working temperature of 40˚C. The HPLC method used was modified from [28]. Lipids were quantified on a Waters 2424 ELS detector based on the peak area in the chromatograms and converted to absolute amounts based on standard curves. The detector settings were as follows: Gain 100, nebulizer heating level set to 30%, drift tube temperature set to 45˚C and pressure set to 40 PSI. The total run time was 41 minutes, using the gradient profile in S1 Appendix Solvent gradient. All lipid analyses including fatty acids and lipid classes were performed using five replicates.

## Amino acid analysis and protein in feed

3 x 100 mg prepared feed was mixed with 0.7 ml milli-Q rinsed water and 0.5 ml 20 mM norleucine (internal standard, Sigma Aldrich). Equal amounts (1.2 ml) 12 M HCl (Sigma Aldrich)

was added to a final concentration of 6 M HCl. Samples were flushed with nitrogen for 10–15 seconds to minimise oxidation, followed by hydrolysis at 110˚C for 22–24 h, according to Moore and Stein [29]. Cooled samples were centrifuged (4000 g) or filtered (0.2 μm) to remove suspended particles. 100 μL sample was evaporated to dryness under nitrogen and resuspended in 1 ml lithium citrate buffer (pH 2.2, Biochrom Co., Cambridge, UK). Samples were stored at 4˚C until analysis. Amino acid samples were analysed chromatographically using a Biochrom 30 Amino acid analyser (Biochrom Co., Cambridge, UK) fitted with a lithium citrate equilibrated ion exchange column and post column derivatisation with ninhydrin. UV signals were analysed using Chromeleon Software (Dionex, Sunnyvale, CA, USA) and identification of amino acid residues was done by using the A9906 physiological amino acid standard (Sigma Aldrich) as described in Mæhre,et al. [30]. Tryptophan is decomposed during acid hydrolysis, and thus cannot be detected in this analysis. Asparagine and glutamine are converted into aspartic and glutamic acid, respectively. Protein content was determined as the sum of individual amino acid residues (the molecular mass of each amino acid with the weight of water subtracted), as recommended by FAO [31].

## Analysis of PUAs

Frozen salmon skin samples (2–3 g) were thawed, extracted in 5 ml 96% aqueous ethanol containing 0.05 mg ml$^{-1}$ butylated hydroxytoluene (BHT, Sigma Aldrich) for 2 hours at room temperature, and centrifuged at 3500 g for 10 minutes. One ml of the supernatant was collected and dinitrophenylhydrazone (DNPH, Sigma Aldrich) derivatives of the aldehydes were prepared by adding 1 ml DNPH solution (0.05M 2,4-dinitrophenyl-hydrazine in acetonitrile:acetic acid 9:1, *v: v*) and incubating at 60˚C for two hours. After the derivatization was completed, 2 ml of water was added, the sample was vortexed for 30 seconds, and the DNPH derivatives were extracted with 4 × 1 ml hexane. The combined hexane phase was reduced to dryness at reduced pressure in a SpeedVac centrifuge, and the sample was reconstituted in 100 μl methanol. Derivatized PUAs were analysed by high resolution mass spectrometry on a Waters Vion IMS qTOF (Milford, MA, USA) coupled to a Acquity I-class UPLC (Waters) with negative electrospray ionization. Chromatographic separation was achieved on a Waters BEH C18 1.7 μm (2.1 × 100 mm) column using water and acetonitrile (both containing 0.1% formic acid) as mobile phases.

## Statistical analysis

Values of the dependent variables are reported as means ± s.e., and all P-values < .05 were considered statistically significant. All data on fish growth were checked for homogeneity of variance and normality by visual inspection by boxplots and histograms. In addition, Lillefors test and Levenes test were used to check for normality and homogeneity respectively. Data were analyzed with the statistical software program R 3.4.0 [32] with the car [33] package and codes from Schlegel [34] and Statistica. All data were normally distributed but had heteroscedastic tendencies. Data from the triplicate tanks were nested under dietary treatment and effects on different responses (SGR, weight, length, condition factor) were analyzed using ANOVA with heteroscedasticity-corrected covariance ("White-Huber covariance") [35–37] and differences between groups were checked with Games-Howell post-hoc test [38]. Data are visualized using package *ggplot2* [39] and Statistica. Simple linear correlation analysis (Pearson r) was performed on lice infestation data and lipid, fatty acids and amino acids in the feed types and fatty acids in salmon fillet.

## Ethics statement

The experiment described has been approved by the local responsible laboratory animal science specialist under the surveillance of the Norwegian Animal Research Authority (NARA)

and registered by the Authority and thereby conforming to Directive 2010/63/EU. The fish was euthanized with overdose benzoak also approved by the local responsible laboratory animal science specialist under the surveillance of the Norwegian Animal Research Authority.

## Results

### Growth of experimental fish and feeding

In total the salmon grew from an initial average mean (±SD) of 134.9 ± 15.4 g to 340.2 ± 49.2 g (n = 291) in 67 days, with an average SGR of 1.4 ± 0.2% day$^{-1}$. SGR was highest between day 34 and day 67 (T1 –T2). From day 67 (T2) to day 82 (T3, end of experiment and lice counting) mean weight had increased to 352.9 ±52.8 g. The modest growth between T2 (day 67) and T3 (day 82) was due to that this was the lice infestation period when fish were kept on a maintenance diet with control (standard) feed. Fish fed diets A and D had overall significantly higher SGR (nested ANOVA, p < 0.001) than fish fed diet B and C for T0 to T3, i.e. for whole period, days 0–67) (Table 2, Fig 1), while SGR for fish fed diet B and C and A and D respectively did not differ statistically. However mean SGR for diet B-Fish oil was lower than for C-*Calanus*.

### Lice infestation

At the end of the experiment (T3, day 82) lice infestation was statistically significantly lower in the salmon group that had been fed diet A-Algae (mean 10.2) than for diets B-Fish oil, C-*Calanus* and especially D-Control (mean 12.9, Fig 2). However, fish fed diets B and C and B and D were statistically indistinguishable.

### Chemical composition of diets

**Total lipid, protein and lipid class composition.** Statistically judged total lipid was similar for all four diets, i.e. ca. 23% of ash-free dry weight (Table 3), while protein content (41.2%) and ash were slightly higher in diet A-Algae than in the other diets that were statistically similar.

Diets A-Algae and B-Fish oil had similar amounts of triacylglycerol (TAG, i.e. ca. 88%), while C-*Calanus* had significantly less (77.28%) and D most (92.94%, Table 4). Diet A, B and D had the same low amounts of wax esters (WE) (1.1–1.3%) contrary to diet C that had ten times more (12.26%). Diacylglycerol (DAG) and free fatty acids (FFA) were comparable for Diets A, B and C (5.19–5.93% and 4.69%), while Diet D had significantly lower values of DAG and FFA (3.49 and 2.41%).

**Fatty acid, protein and amino acid composition.** The omega-3 fatty acids EPA (C20:5n-3) and DHA (C22:6n-3) were present in comparable amounts in diets A, B and C (16.5–16.9%), and less than half of this (7.4%) was in diet D (Table 5). DHA was present in slightly larger amounts than EPA in all diets. Palmitic acid (C16:0), stearidonic acid (C18:4n-3), cetoleic acid (C22:1n-11) and elaidic acid (C18:1n-9) dominated and were present in comparable amounts in diets A, B and C. Palmitic acid (C16:1n-7), stearidonic acid (C18:4n-3) and cetoleic

**Table 2. Specific salmon growth rate (% day$^{-1}$) during the experimental periods.**

| Diet | A-Algae | | B-Fish oil | | C-*Calanus* | | D-Control | |
|---|---|---|---|---|---|---|---|---|
| | Mean | ± S.D. | Mean | ± S.D. | Mean | ± S.D. | Mean | ± S.D. |
| **Days 0–34 (T0-T1)** | 1.18 | 0.03 | 1.06 | 0.05 | 1.01 | 0.04 | 1.20 | 0.04 |
| **Days 34–67 (T1-T2)** | 1.67 | 0.01 | 1.52 | 0.02 | 1.70 | 0.01 | 1.69 | 0.02 |
| **Days 0–67 (T0-T3)** | 1.42 | 0.02 | 1.29 | 0.03 | 1.35 | 0.02 | 1.44 | 0.02 |

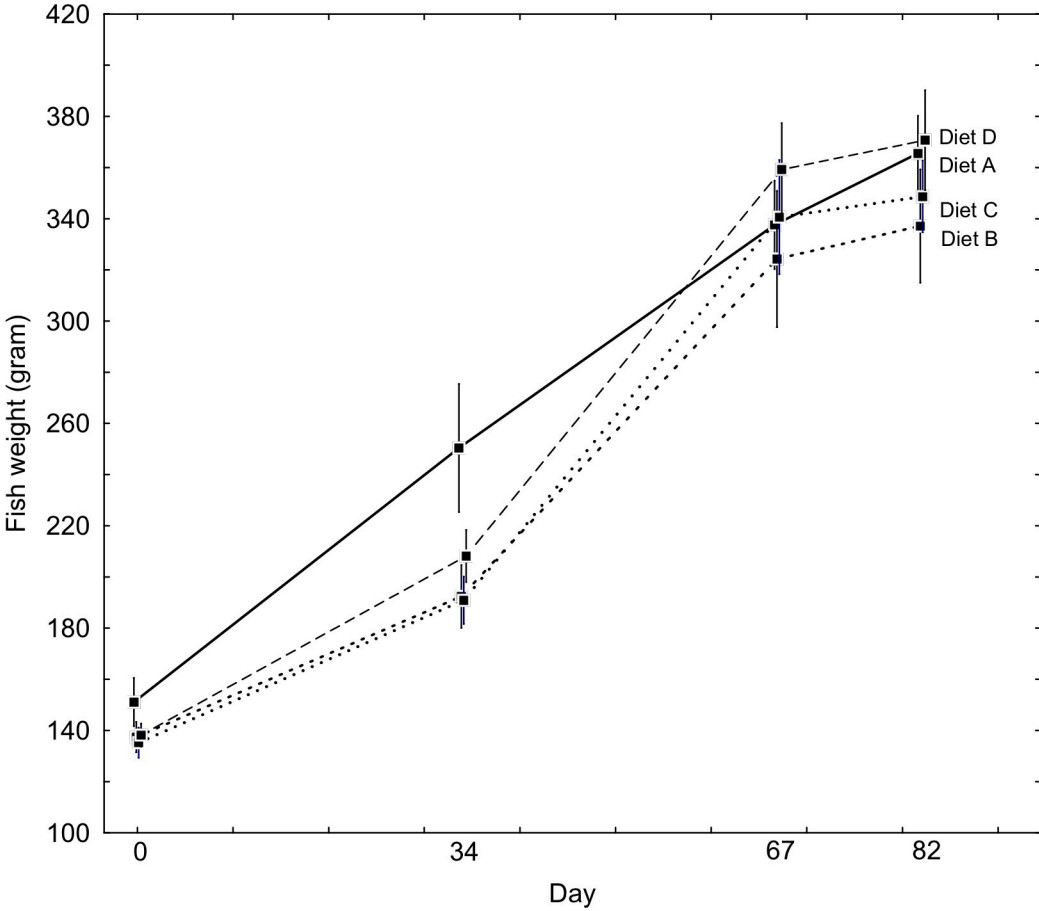

**Fig 1. Weight vs. time for salmon fed the different (A–D) diets.**

acid (C22:1n-11) were present at significantly lower amounts (ca. 50%) and elaidic acid at higher amounts in diet D. Further SFA (Saturated fatty acids) was comparable in diets A, B and C (20,7–21.2%) and much lower in diet D (12.8%), while MUFAs (Monunsaturated fatty acids) and PUFAs (Polyunsaturated fatty acids) were present in reasonably comparable amounts (ranges 42.7–52.2% and 33.9–36.6%). There was negligible difference in the protein and amino acid content and composition between the different diets (Tables 3 and 6). One-way ANOVA though revealed some small but statistically significant ($p < 0.05$) differences. Diet A (Algae) and C (*Calanus*) were identical with respect to amino acid content, except for glutamic acid (Glu) that was significantly lower in diet A than in the other diets. The same was the case for valine in A. Further proline was significantly lower in D than in the other diets. The largest difference was though for Tyrosine between diet B (11.1 mg) and the other diets (12.4–13.6 mg).

   **Protein, lipid, lipid class and fatty acid composition in salmon.** The protein content of salmon flesh varied around 15% of wet weight and there were no statistically significant (one—way Anova) differences between the experimental groups (Table 7). Further there was no statistical differences between diets B, C and D (4.3, 4.5, 4.3% of WW) while salmon fed the algae (A) diet had significantly lower total lipid content (3.7%). The fatty acid composition of the salmon flesh varied considerably after 67 days of experimental feeding (Table 8). All 3 groups fed diets with high content of marine lipids (A, B and C) contained less LA (18:2n-6)

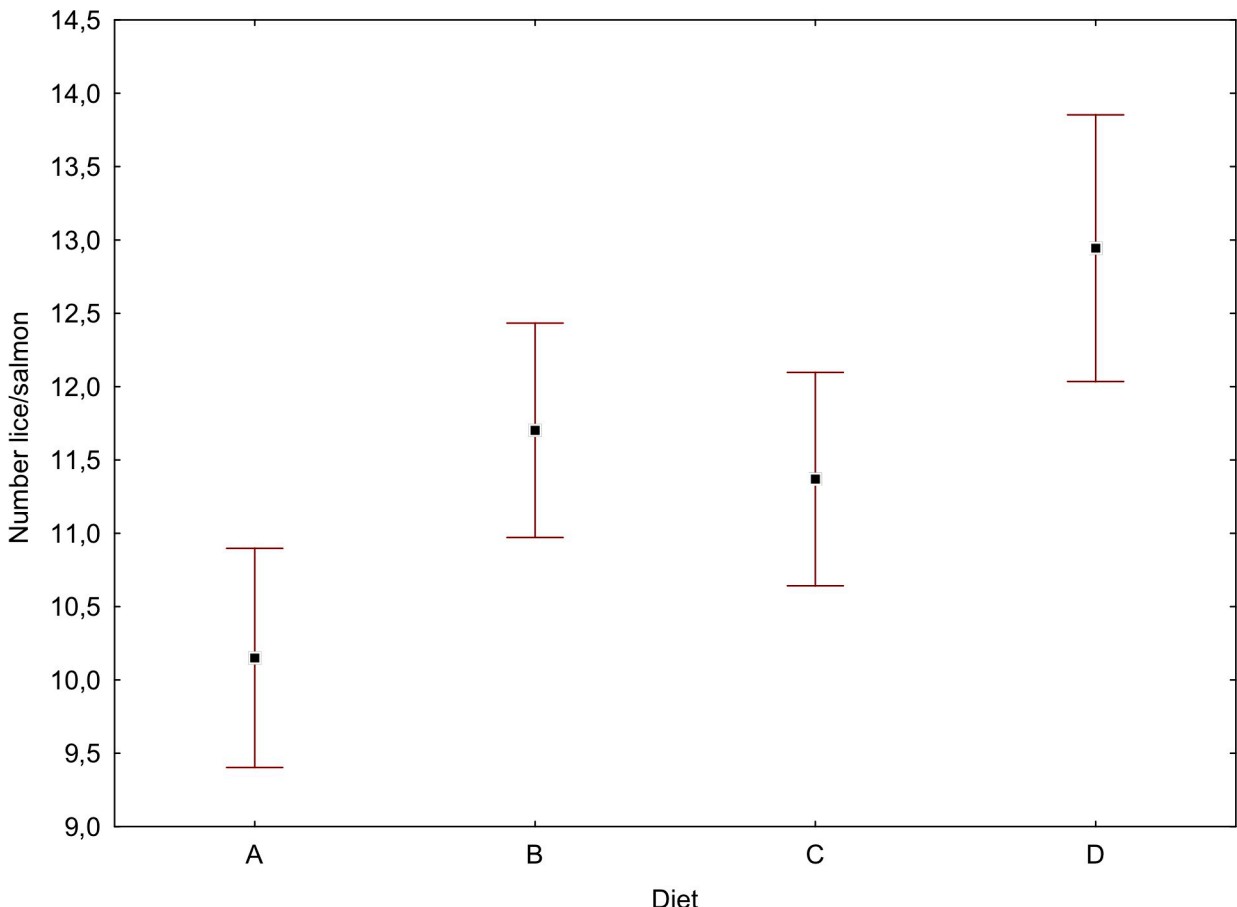

**Fig 2. Number of lice per fish fed diets A–D on experiment day 82 (T3).**

and ALA (18:3n-3) compared to fish fed diet D. Fish fed diet B were highest in omega-3 PUFAs (EPA + DHA), i.e. 0.64g/100g flesh, followed by diet A (0.50), D (0.40) and C (0.19). Diet C resulted in the lowest amounts of omega-3 PUFAs, i.e. less than half of the content in the other treatments. Fish fed diet A had intermediate levels of all these n-3 PUFAs. Most of the MUFAs, i.e. 16:1n-7, 20:1n-9, 22:1n-11, differed significantly, and 22:1n-9 were lowest in D whereas 18:1n-9 was highest in D.

## Discussion

In the present study, fish fed diet A-Algae feed had statistically significant fewer lice (Fig 2) compared to fish fed the other diet types. The reasons for this may be: i) Differences in general health condition or stress level of salmon groups prior to exposure to salmon lice copepodites

**Table 3. Water, protein (sum of amino acid residues), ash and total lipid content as % of DW in the different diets, n = 5.**

| Diet | A-Algae | | B-Fish oil | | C-*Calanus* | | D-Control | |
|---|---|---|---|---|---|---|---|---|
| | Mean | ± S.D. | Mean | ± S.D. | Mean | ± S.D. | Mean | ± S.D. |
| **Water** | 8.0 | 0.1 | 9.8 | 0.0 | 9.3 | 0.0 | 8.6 | 0.0 |
| **Protein** | 41.2 | 0.2 | 40.0 | 0.2 | 40.3 | 0.6 | 40.3 | 0.1 |
| **Total lipids** | 23.4 | 1.05 | 23.3 | 0.78 | 23.7 | 0.14 | 23.2 | 0.6 |
| **Ash** | 9.4 | 0.0 | 8.2 | 0.0 | 8.3 | 0.0 | 8.3 | 0.0 |

**Table 4. Lipid class composition of diets as weight % of total lipid, triacylglycerol (TAG), wax esters (WE), diacylglycerol (DAG), free fatty acids (FFA).**

| Diet | A-Algae | | B-Fish oil | | C-*Calanus* | | D-Control | |
|---|---|---|---|---|---|---|---|---|
| | Mean | ± S.D. | Mean | ± S.D. | Mean | ± S.D. | Mean | ± S.D. |
| TAG | 87.71 | 0.85 | 88.51 | 1.56 | 77.28 | 2.14 | 92.94 | 0.22 |
| WE | 1.15 | 0.17 | 1.28 | 0.14 | 12.26 | 1.50 | 1.17 | 0.07 |
| DAG | 5.93 | 0.38 | 5.52 | 0.56 | 5.19 | 0.47 | 3.49 | 0.28 |
| FFA | 5.21 | 0.37 | 4.69 | 0.93 | 5.28 | 0.32 | 2.41 | 0.20 |

may have influenced rate and magnitude of infestation [4, 5]; ii) Variable health may have resulted from differences in chemical composition between the diets; iii) An ingredient present in the diet, may have triggered production of some bioactive compound that influenced lice

**Table 5. Fatty acid content of diets as weight % of total lipid, n = 5, SFA = Saturated fatty acids, MUFA = Monunsaturated fatty acids, PUFA = Polyunsaturated fatty acids.**

| Diet | A-Algae | | B-Fish oil | | C-*Calanus* | | D-Control | |
|---|---|---|---|---|---|---|---|---|
| | Mean | ± S.D. | Mean | ± S.D. | Mean | ± S.D. | Mean | ± S.D. |
| C12:0 | 0.0 | 0.00 | 0.0 | 0.00 | 0.0 | 0.00 | 0.0 | 0.00 |
| C14:0 | 5.7 | 0.03 | 5.7 | 0.03 | 6.5 | 0.17 | 2.4 | 0.04 |
| C14:1 | 0.3 | 0.00 | 0.3 | 0.00 | 0.3 | 0.01 | 0.0 | 0.00 |
| C16:0 | 13.1 | 0.03 | 13.2 | 0.02 | 12.9 | 0.09 | 8.5 | 0.04 |
| C16:1n-7 | 6.4 | 0.02 | 6.3 | 0.03 | 6.1 | 0.02 | 2.8 | 0.03 |
| C16:2n-4 | 0.4 | 0.01 | 0.4 | 0.00 | 0.4 | 0.00 | 0.2 | 0.00 |
| C16:3n-4 | 0.3 | 0.00 | 0.2 | 0.00 | 0.2 | 0.01 | 0.0 | 0.00 |
| C18:0 | 1.7 | 0.00 | 1.7 | 0.00 | 1.7 | 0.01 | 1.6 | 0.05 |
| C18:1n-9 | 13.0 | 0.01 | 12.9 | 0.01 | 13.0 | 0.09 | 37.6 | 0.23 |
| C18:1n-7 | 2.7 | 0.02 | 2.7 | 0.01 | 2.6 | 0.03 | 3.0 | 0.04 |
| C18:2n-6 | 4.2 | 0.04 | 4.8 | 0.15 | 4.9 | 0.06 | 14.3 | 0.10 |
| C18:3n-6 | 0.0 | 0.00 | 0.0 | 0.00 | 0.0 | 0.00 | 0.0 | 0.00 |
| C18:3n-4 | 0.0 | 0.00 | 0.0 | 0.00 | 0.0 | 0.00 | 0.0 | 0.00 |
| C18:3n-3 | 1.3 | 0.01 | 1.3 | 0.01 | 1.5 | 0.03 | 6.8 | 0.04 |
| C20:0 | 0.2 | 0.00 | 0.2 | 0.01 | 0.2 | 0.01 | 0.4 | 0.00 |
| C20:1n-9 | 3.4 | 0.02 | 3.4 | 0.02 | 6.7 | 3.73 | 1.4 | 0.03 |
| C18:4n-3 | 11.6 | 0.05 | 11.5 | 0.04 | 8.1 | 3.78 | 5.5 | 0.06 |
| C20:2n-6 | 0.3 | 0.01 | 0.3 | 0.00 | 0.3 | 0.01 | 0.2 | 0.04 |
| C20:3n-6 | 0.0 | 0.00 | 0.0 | 0.00 | 0.0 | 0.00 | 0.0 | 0.00 |
| C20:4n-6 | 0.4 | 0.01 | 0.4 | 0.01 | 0.4 | 0.01 | 0.2 | 0.01 |
| C20:3n-3 | 0.0 | 0.00 | 0.0 | 0.00 | 0.0 | 0.00 | 0.2 | 0.01 |
| C22:1n-11 | 14.5 | 0.07 | 14.3 | 0.04 | 13.5 | 0.11 | 5.9 | 0.08 |
| C22:1n-9 | 1.9 | 0.03 | 1.8 | 0.02 | 1.9 | 0.08 | 0.9 | 0.02 |
| C20:5n-3 | 7.3 | 0.05 | 7.1 | 0.03 | 7.6 | 0.05 | 3.1 | 0.05 |
| C22:4n-6 | 0.3 | 0.00 | 0.3 | 0.00 | 0.3 | 0.01 | 0.0 | 0.07 |
| C24:1n-9 | 0.9 | 0.01 | 0.9 | 0.00 | 0.9 | 0.02 | 0.5 | 0.00 |
| C22:5n-3 | 0.8 | 0.02 | 0.8 | 0.01 | 0.8 | 0.03 | 0.3 | 0.06 |
| C22:6n-3 | 9.4 | 0.06 | 9.4 | 0.03 | 9.4 | 0.06 | 4.2 | 0.07 |
| Σ SFA | 20.7 | 0.05 | 20.8 | 0.04 | 21.2 | 0.19 | 12.8 | 0.07 |
| Σ MUFA | 43.1 | 0.08 | 42.7 | 0.06 | 44.9 | 3.73 | 52.2 | 0.25 |
| Σ PUFA | 36.2 | 0.11 | 36.6 | 0.16 | 33.9 | 3.78 | 34.9 | 0.18 |
| EPA+DHA | 16.7 | 0.1 | 16.5 | 0.0 | 16.9 | 0.1 | 7.4 | 0.1 |

**Table 6. Essential Amino acids (EAA, mg gDW⁻¹), Arginine = Arg, Histidine = His, Isoleucine = Ile, Leucine = Leu, Lysine Lys, Methionine = Met, Phenylalanine = Phe, Threonine = Thr, Tryptophan = Trp, Valine = Val; Nonessential ones (NEEA): Alanine = Ala, Asparagine = Asn, Aspartic acid = Asp, Cysteine = Cys, Glycine = Gly, Glutamic acid = Glu, Glutamine = Gln, Proline = Pro, Serine = Ser, Tyrosine = Tyr, Hydroxyprolin = Hyp.Taurin = Tau, TAA = Total amino acids (sum of amino acid residues), %EAA = Essential amino acids as % of total.**

| Diet | Amino acid | A-Algae | | B-Fish oil | | C-*Calanus* | | D-Control | |
|---|---|---|---|---|---|---|---|---|---|
| | | Mean | ± S.D. | Mean | ± S.D. | Mean | ± S.D. | Mean | ± S.D. |
| **EAA** | Thr | 18.0 | 0.0 | 18.1 | 0.0 | 18.1 | 0.3 | 17.9 | 0.2 |
| | Val | 19.0 | 0.2 | 19.6 | 0.1 | 19.5 | 0.3 | 19.7 | 0.1 |
| | Met | 10.2 | 0.1 | 10.7 | 0.1 | 10.3 | 0.1 | 10.5 | 0.1 |
| | Ile | 17.6 | 0.1 | 17.9 | 0.2 | 17.9 | 0.2 | 17.9 | 0.2 |
| | Leu | 35.9 | 0.1 | 36.5 | 0.2 | 36.6 | 0.4 | 36.3 | 0.2 |
| | Phe | 22.4 | 0.0 | 22.8 | 0.2 | 22.8 | 0.2 | 22.5 | 0.2 |
| | Lys | 30.9 | 0.1 | 31.3 | 0.2 | 31.4 | 0.4 | 31.3 | 0.2 |
| | His | 8.9 | 0.1 | 9.2 | 0.0 | 9.2 | 0.1 | 9.0 | 0.1 |
| | Trp | | | | | | | | |
| **NEAA** | Asp | 28.8 | 0.1 | 28.9 | 0.1 | 28.9 | 0.4 | 28.6 | 0.2 |
| | Ser | 23.5 | 0.0 | 23.7 | 0.1 | 23.9 | 0.3 | 23.6 | 0.2 |
| | Glu | 115.7 | 0.8 | 119.5 | 1.0 | 119.1 | 1.3 | 117.8 | 0.7 |
| | Pro | 40.9 | 0.3 | 41.2 | 0.5 | 40.3 | 0.6 | 39.1 | 0.3 |
| | Gly | 25.4 | 0.2 | 25.5 | 0.1 | 25.4 | 0.2 | 25.0 | 0.1 |
| | Ala | 23.1 | 0.2 | 23.0 | 0.0 | 23.2 | 0.3 | 23.0 | 0.1 |
| | Cys | 5.6 | 0.1 | 5.6 | 0.1 | 5.7 | 0.1 | 5.8 | 0.1 |
| | Tyr | 13.3 | 0.4 | 11.1 | 0.9 | 12.4 | 1.4 | 13.6 | 0.8 |
| | Arg | 29.1 | 0.6 | 29.1 | 0.5 | 28.5 | 0.8 | 30.5 | 0.1 |
| | TAA | 468.2 | 2.3 | 473.6 | 3.0 | 473.2 | 7.6 | 472.0 | 1.6 |
| | EAA | 34.8 | 0.1 | 35.1 | 0.1% | 35.0 | 0.2 | 35.0 | 0.3 |
| **Other** | Hyp | 2.6 | 0.1 | 2.5 | 0.1 | 2.7 | 0.2 | 2.3 | 0.1 |
| | Tau | 1.4 | 0.0 | 1.4 | 0.0 | 1.4 | 0.0 | 1.3 | 0.0 |

infestation, i.e. in accordance with our original hypothesis where the active agents were PUAs [20].

Farmed fish needs lipid, protein, energy, vitamins and minerals to thrive and develop naturally. Here lipids and especially essential fatty acids of marine origin are important for normal physiological functions [40]. Salmon in our experiment had specific growth rates from 1.29 to 1.44% day⁻¹ during the period (67 days) they were fed the four diets. In a study where seven different diets with variable levels of animal by-products, vegetable proteins, fish oil and rapeseed oil were used to feed Atlantic salmon, obtained SGR values were maximum 0.65% day⁻¹ for 14 weeks [41]. In other comparable tank and pen studies with Atlantic salmon at temperature regimes around 10–11 °C or lower, maximum growth rates were 0.94–1.17% day⁻¹ [42–44]. All our salmon groups showed good growth for all diets (mean

**Table 7. Water, lipid and protein composition (% of wet weight) of Atlantic salmon (*Salmo salar* L) juveniles fed diet A, B, C and D (n = 9 for each diet).**

| Diet | A-Algae | | B-Fish oil | | C-*Calanus* | | D-Control | |
|---|---|---|---|---|---|---|---|---|
| | Mean | ± S.D. | Mean | ± S.D. | Mean | ± S.D. | Mean | ± S.D. |
| **Water** | 75.7 | 0.5 | 75.5 | 0.8 | 74.6 | 0.4 | 71.1 | 1.3 |
| **Protein** | 15.6 | 0.4 | 15.4 | 0.5 | 14.9 | 0.2 | 15.2 | 0.3 |
| **Total lipids** | 3.7 | 0.4 | 4.3 | 0.5 | 4.5 | 0.5 | 4.3 | 0.5 |
| **Ash** | 1.2 | 0.0 | 1.3 | 0.1 | 1.4 | 0.1 | 1.2 | 0.1 |

**Table 8. Amount of fatty acid (g per 100g of flesh) in Atlantic salmon juveniles (*Salmo salar L.*) fed diet A, B, C and D after 67 days of feeding, n = 9.**

| Diet | A-Algae | | B-Fish oil | | C-*Calanus* | | D-Control | |
|---|---|---|---|---|---|---|---|---|
| | Mean | ± S.D. | Mean | ± S.D. | Mean | ± S.D. | Mean | ± S.D. |
| C14:0 | 0.14 | 0.02 | 0.15 | 0.02 | 0.18 | 0.02 | 0.09 | 0.01 |
| C16:0 | 0.52 | 0.06 | 0.56 | 0.07 | 0.60 | 0.15 | 0.48 | 0.05 |
| C16:1n-7 | 0.12 | 0.01 | 0.17 | 0.03 | 0.17 | 0.02 | 0.09 | 0.01 |
| C18:0 | 0.11 | 0.01 | 0.11 | 0.01 | 0.12 | 0.01 | 0.12 | 0.01 |
| C18:1n-12 | 0.05 | 0.00 | 0.05 | 0.01 | 0.06 | 0.01 | 0.04 | 0.01 |
| C18:1n-9 | 0.49 | 0.06 | 0.64 | 0.10 | 0.66 | 0.07 | 1.15 | 0.13 |
| C18:1n-7 | 0.09 | 0.01 | 0.11 | 0.02 | 0.11 | 0.01 | 0.11 | 0.01 |
| C18:2n-6 | 0.16 | 0.02 | 0.22 | 0.04 | 0.22 | 0.02 | 0.39 | 0.04 |
| C18:3n-3 | 0.04 | 0.01 | 0.07 | 0.01 | 0.07 | 0.01 | 0.16 | 0.02 |
| C18:4n-3 | 0.07 | 0.01 | 0.09 | 0.01 | 0.10 | 0.01 | 0.05 | 0.01 |
| C20:1n-9 | 0.27 | 0.03 | 0.34 | 0.05 | 0.36 | 0.04 | 0.21 | 0.02 |
| C20:2n-6 | 0.02 | 0.00 | 0.03 | 0.00 | 0.03 | 0.00 | 0.04 | 0.00 |
| C20:4n-6 | 0.02 | 0.01 | 0.01 | 0.00 | 0.01 | 0.00 | 0.01 | 0.00 |
| C22:1n-11 | 0.29 | 0.03 | 0.35 | 0.05 | 0.38 | 0.04 | 0.18 | 0.02 |
| C22:1n-9 | 0.07 | 0.01 | 0.09 | 0.01 | 0.09 | 0.01 | 0.05 | 0.01 |
| C20:5n-3 | 0.12 | 0.01 | 0.16 | 0.02 | 0.16 | 0.02 | 0.08 | 0.01 |
| C24:1n-9 | 0.03 | 0.00 | 0.03 | 0.00 | 0.03 | 0.00 | 0.02 | 0.00 |
| C22:5n-3 | 0.05 | 0.00 | 0.06 | 0.01 | 0.03 | 0.01 | 0.03 | 0.00 |
| C22:6n-3 | 0.38 | 0.03 | 0.48 | 0.05 | 0.03 | 0.02 | 0.32 | 0.03 |
| Σ SFA | 0.77 | 0.09 | 0.83 | 0.11 | 0.89 | 0.09 | 0.68 | 0.08 |
| Σ MUFA | 1.39 | 0.14 | 1.78 | 0.27 | 1.86 | 0.20 | 1.86 | 0.21 |
| Σ PUFA | 0.88 | 0.07 | 1.13 | 0.14 | 1.12 | 0.11 | 1.09 | 0.11 |
| EPA + DHA | 0.50 | 0.04 | 0.64 | 0.06 | 0.19 | 0.03 | 0.40 | 0.04 |
| n-3/n-6 | 3.3 | | 3.3 | | 1.5 | | 1.45 | |

SGR 1.4 ± 0.2% day$^{-1}$). Our fish hence performed in the high end of obtainable growth for salmon at the same weight and temperature [45]. The SGR range between the four diet groups also varied little (1.29 to 1.44% day$^{-1}$), and we interpret this as an indication of that the fish were at acceptable health compared to what is common in salmon aquaculture. In the present study all four diets contained 25% fish meal. In addition, diets A, B and C contained ca. 21.3% oils of marine origin while diet D had 9% marine and 12.3% rapeseed oil (Table 1). In the study by [41], they applied comparable amounts of fish meal except for one out of seven diets, but all diets had substantially lower quantities of marine oils compared to our 20%. They observed the best growth with the diet that had the largest amounts of fish meal and marine oil, but growth was unaffected as long as fish oil and fish meal were above 5% respectively. Further growth was correlated to total omega-3 fatty acid content in the diets, this irrespectively of the content of individual omega-3 fatty acids. Other investigations where fish oil inclusion in the feed was lower than in our study, showed substantially or slightly lower growth rates [46, 47]. The main reason for the good growth performance in our experiment was therefore probably due to the content of marine ingredients. In 2016 the overall content of marine and terrestrial plant ingredients in Norwegian aquaculture was 25% and 71% [48]. Our diet with the lowest inclusion of marine food thus matched this and confirms that the composition of our diets represents normal or high inclusion of marine biomass relative to common aquaculture procedure, and that this indicates that our salmon was fed diets that supported good health.

As stated fish fed diet A-Algae had statistically significant fewer lice (Fig 2) compared to fish fed the other feed types. We hold it unlikely that this was caused by some general effect of fish health and growth, this since fish fed all diets performed well and hence appeared to be at good health throughout the experiment. Salmon fed diets A-Algae and D-Control had slightly better growth than diets B-Fish oil and C-*Calanus* (Fig 1, Table 2). Even if these differences were low, they were statistically significant. The reason for this difference must therefore be sought in some common denominator in the diets A and D vs. B and C, and if this could have both caused the reduced lice infestation with diet A and the higher amounts of lice with diet D (lice in group B and C were statistically indistinguishable). In general, there was little variance in the fatty acids and amino acids between the diets and for FAs in fish flesh. A notable statistically significant deviation was that diet D had high amounts of the MUFA C18:1n-9, linoleic acid (C18:2n-6) and ALA (C18:3n-3). The same tendency, though weaker, was present for the same FAs in the flesh of fish fed diet D. This demonstrates that for these FAs the feed composition influences fish flesh, but this by no way correlated with the distribution of lice between the experimental groups. Diet D was also low in EPA and DHA (7.4%) relative to diets A, B and C (16.5–16.9%). Free fatty acids (FFA) were also low (2.41%), relative to the other diets (4.69–5.28%). If salmon lice infection was in some way related to altered fatty acid composition in the feed, i.e. more specifically omega-3, this fails to explain the gradient in lice infestation in salmon fed diets A, B and C.

Our original lice-hypothesis was based on that polyunsaturated aldehydes (PUAs), e.g. 2-trans, 4-trans decadenial (A3) produced by diatoms can function as grazing deterrents and harm copepod development. The formation of FFA in diatoms can be attributed to grazer defense mechanisms, where the fatty acids are released from the glycerol backbones via hydrolytic enzymes to produce oxylipins downstream. We were not able to detect any meaningful and significant correlation here between diet composition, fillet composition and lice amount. The most notable result, "close to significant", was though that salmon fed diet D was lowest in FFA and diets A and C highest.

If diatoms are the original producer of an aldehyde functioning as a lice deterrent in wild salmon, it is to be expected that it is fed up the food web since zooplankton and small fish are intermediates between primary producers and salmon. Our lice infestation mean values were positively correlated to trophic level, i.e the number of lice increased from diatom feed through *Calanus*, fish oil to standard feed that had terrestrial ingredients. This may indicate that even if we were not able to detect it, that it was some ingredient/bioactive in the feed that caused variable lice attacks. Skin samples from individuals belonging to the four different feed regimes were analyzed for contents of PUAs using a commercially available standard of 2-*trans*-4-*trans*-decadienal (DD) as reference. Prior to analysis, DNPH derivatives of the PUAs were in order to improve their stability and make them more amenable for analysis on LC-MS. The derivatized standard of DD was detected as a peak at $R_t$ 10,4 minutes and m/z 331,1420 ($[M-H]^-$). This signal, indicating presence of DD, could not be detected in any of the salmon skin samples. Diatom lipids are often highly unsaturated, and this makes them easily prone to oxidation. The α,β,γ,∂-unsaturated aldehyde 2-*trans*-4-*trans*-decadienal (DD) was together with two other polyunsaturated aldehydes (PUAs) first identified in the diatoms *Skeletonema costatum*, *Pseudo-nitzschia delicatissima* and *Thalassiosira rotula* [49–51]. The production of PUAs in diatoms is an enzymatic process where PUFAs are initially released from membrane lipid storages by A2—phospholipase, and the free PUFAs are sequentially transformed into lipid hydroperoxides and PUAs by lipoxygenases and hydroxyperoxide lyases, respectively [20, 52, 53]. The seabird crested auklet (*Aethia cristatella*) secretes lipids to the plumage, and this secretion contain aldehydes that have been shown to act as invertebrate repellents as they paralyze lice [54]. However, the reportoir of PUAs that can be produced by any organism is

potentially large as it depends not only on the fatty acid profile of the membrane lipids, but also to the variety of the pool of relevant enzymes that is available for PUA production [55]. FFA was high in diet A-Algae and C (*Calanus*), lower in diet B (fish oil) and lowest in diet D (Control), i.e. negatively related to lice infestation. FFA can act as a proxy for oxidative stress, and this may support the hypothesis that the lice deterring agent is a PUA and in fat produced in/by the salmon.

## Conclusion

Our investigation showed statistically significantly reduced lice infestation in salmon offered conventional feed with diatom biomass. In our investigation there were no correlations between feed ingredients and lice infestation. Neither were we able to detect a bioactive compound that was active as a lice deterrent, nor could we relate differences in lice infestation to health differences. The main result that lice infestation was reduced in the salmon group offered diatom biomass still indicates the presence of some active lice deterring ingredient. In addition to that the ingredient may come from the feed, it can also have been synthesized or resulted from chemical reactions in the fish. We therefore strongly recommend that further investigations should be undertaken in order to get more insight into the potential production and secretion of invertebrate repelling aldehydes in salmon, and that experiments should also take place in open sea pens under natural conditions".

## Supporting information

**S1 Appendix. Solvent gradient.**
(DOCX)

**S1 File. Fish growth and lice.**
(XLSX)

**S2 File. Lipid classed feed.**
(XLS)

**S3 File. Fatty acids feed.**
(XLSX)

**S4 File. Amino acids feed.**
(XLSX)

**S5 File. DW etc feed.**
(XLSX)

**S6 File. Total lipid feed.**
(XLSX)

## Acknowledgments

We want to thank Gunilla K. Eriksen, Jo Strømholt and John-Steinar Bergum at the ferrosilicon producer Finnfjord AS for valuable help during production of the diatom biomass.

## Author Contributions

**Conceptualization:** Hans Chr. Eilertsen, Birthe Vang, Sten Siikavuopio.

**Data curation:** Hans Chr. Eilertsen, Ingeborg Hulda Giæver, Jon Brage Svenning, Lars Dalheim, Ragnhild Aven Svalheim, Sten Siikavuopio, Ragnhild Dragøy, Espen Hansen, Anette Hustad, Karl-Erik Eilertsen.

**Methodology:** Edel Elvevoll, Ragnhild Dragøy, Richard A. Ingebrigtsen.

**Validation:** Hans Chr. Eilertsen.

**Writing – review & editing:** Hans Chr. Eilertsen, Espen Hansen.

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
