## [Decision Letter · Decision Letter 0]

19 Apr 2021

PONE-D-21-08390

Inclusion of photoautotrophic cultivated diatom biomass in salmon feed deters lice

PLOS ONE

Dear Dr. Eilertsen,

Thank you for submitting your manuscript to PLOS ONE. After careful consideration, we feel that it has merit but does not fully meet PLOS ONE’s publication criteria as it currently stands. Therefore, we invite you to submit a revised version of the manuscript that addresses the points raised during the review process.

1. The data cannot support the conclusions. PLOS ONE is designed to communicate primary scientific research, and welcome submissions in any applied discipline that will contribute to the base of scientific knowledge. But the data of this manuscript cannot support the conclusions.

2. The revised manuscript needs to address each of the comments of the reviewers.

We look forward to receiving your revised manuscript.

Kind regards,

Tzong-Yueh Chen, Ph.D.

Academic Editor

PLOS ONE

Journal Requirements:

2. Please amend your Ethics Statement to include the following information from your Methods section:

'The experiment described has been approved by the local responsible laboratory animal science specialist under the surveillance of the Norwegian Animal Research Authority (NARA) and registered by the Authority and thereby conforming to Directive 2010/63/EU. The fish were euthanized with overdose benzoak also approved by the local responsible laboratory animal science specialist under the surveillance of the Norwegian Animal Research Authority.'

"NO"

ting Interests on the online submission form to state any Competing Interests. If you have no competing interests, please state "The authors have declared that no competing interests exist.", as detailed online in our guide for authors at http://journals.plos.org/plosone/s/submit-now

4. Please upload a copy of Figures 3 and 4, to which you refer in your text on page 22. If the figure is no longer to be included as part of the submission please remove all reference to it within the text.

5. Please include a caption for figures 3 and 4.

Reviewers' comments:

Reviewer's Responses to Questions

**Comments to the Author**

1. Is the manuscript technically sound, and do the data support the conclusions?

Reviewer #1: Partly

Reviewer #2: Yes

2. Has the statistical analysis been performed appropriately and rigorously? 

Reviewer #1: Yes

Reviewer #2: Yes

3. Have the authors made all data underlying the findings in their manuscript fully available?

Reviewer #1: Yes

Reviewer #2: Yes

4. Is the manuscript presented in an intelligible fashion and written in standard English?

Reviewer #1: Yes

Reviewer #2: Yes

5. Review Comments to the Author

Reviewer #1: The authors evaluated the potential of diatom biomass as a lice-reducing ingredient in salmon feed.

1. How about the dose-dependent manner of the diatom biomass in the present study?

2. The potential indicator or the active compound for reducing lice in diatom were necessary to achieve the reproducibility in the present study.

3. The authors assessed the composition in diets and salmon, and those results were further used to analyze the correlation to lice infestation. Unfortunately, there was no direct linkage between the decreased lice infestation and diatom.

Reviewer #2: This study examined the effects of feeding Atlantic salmon with feed containing different sources of oil from diatom, fish, Calanus sp., and rapeseed on anti lice responses. The authors analysed thoroughly of the amino acid and fatty acid content and classes from each experimental feed as well as from fish flesh. The authors found that fish that fed with feed containing diatom oil had the least number of sea lice among all the groups examined, but they fail to find which substance (bioactive factor) may responsible for this phenomenon.

Some modifications are required.

1. The language of this manuscript should be properly checked by English native speaker. The content in the "Introduction" is a bit too long with redundant setences.

2. "A" to "An" on line 36.

3. "long-chain polyunsaturated fatty acids (LC-PUFA)" to "LC-PUFA" on line 72.

4. What is "temperature areas" in line 102?

5. Please modify paragraph in lines 107-111.

6. The range between 50-70000L is quite huge (lines 125-126), please make sure on this description.

7. Please describe how the standard protocol of exposure was done (lines 169-170).

8. Line 177, "try" to "dry".

9. Line 216, "International" to "international".

10. Please use dot (.) not comma (,) for decimal point through the whole manuscript.

11. The format of describing company is not always the same, please correct it (e.g. see lines 242 and 254 on page 11 for Sigma Aldrich).

12. Line 295, "Statment" to"Statement".

13. The description on lines 311-312 is wrong (e.g. fish fed with diet A and D was significant higher than fish that fed with diet B and C for T1 to T2), please correct it.

14. Please provide the content of carbohydrate in Table 3.

15. The description on lines 366-367 is wrong, please correct it.

16. Line 369, "Tyrosine" to "tyrosine".

17. Line 405, "EEA" to "EAA".

18. Line 422, "were" to "higher" (I am not sure)?

19. Line 453, "[42] [43,44]" to "[42-44]".

20. Lines 455-458. The authors should not summarize higher growth in this manuscript compared to other previous studies is due to good health. Does that mean the fish in other studies are all in bad health?

21. Lines 488 and 490, I can't see Fig.3 and Fig. 4. Are they suppose to be Table 5 and Table 8, respectively?

22. What is "though stands" (Line 544)?

23. Line 545, "healt" to "health".

24. Please organize the format for "Reference"

25. The language in supplementary data should switch to English.

6. PLOS authors have the option to publish the peer review history of their article (what does this mean?). If published, this will include your full peer review and any attached files.

Reviewer #1: No

Reviewer #2: No

---

## [Author Response · Author response to Decision Letter 0]

27 May 2021

To the Academic editor, PLOS ONE 

Tzong-Yueh Chen

 UiT The arctic university of Norway

I hereby submit a revised version of the manuscript no. PONE-D-21-08390 with title “Inclusion of photoautotrophic cultivated diatom biomass in salmon feed deters lice” (new title: “Inclusion of photoautotrophic cultivated diatom biomass in salmon can deter lice”. We have changed the manuscript as follows, according to the points raised by the academic editor and reviewer, plus some other issues. Our response and changes are:

Academic editor comments: Responses

1. The data cannot support the conclusions I guess here is meant that we failed to detect an active ingredient/compound that could explain the reduction in lice infestation. In the original version it was stressed that so was the case, and that the only thing that differed significantly was FFA that was present at higher concentrations in feed with algae (and Calanus). We though feel that it was correct, at this stage, not to attribute this to lice reduction! We have though changed the Conclusion to be more “balanced” (see further below), and the title to “can deter lice” from “deters lice”, plus several changes in the manuscript itself. 

2-5: -Change ethics Statement to include information from Methods section; 

-Please upload a copy of Figures 3 and 4, to which you refer in your text on page 22. If the figure is no longer to be included ….. -Ethics statement included

-No competing interest statement fixed

-Figures 3 and 4 were removed from manuscript, all references to them are now removed!

Reviewers comments: 

Reviewer #1: 

1. How about the dose-dependent manner of the diatom biomass in the present study? This was a pilot study with limited amounts of microalgae (diatom) biomass available. The aim was therefore to monitor growth and lice infestation vs. chemical agents represented by feed with different origins (fish oil, Calanus oil, standard feed) rather than different amounts of diatom biomass. But, an experiment underway will be aimed at examining this, i.e. different amounts of diatom biomass (with all other constituents and treatments kept constant).

2,3 = Remarks that are not questions, from referee 

Reviewer #2: 

1. The language of this manuscript should be properly checked by English native speaker. The content in the "Introduction" is a bit too long with redundant sentences. Language is checked and Introduction has been shortened! See file «Revised Manuscript with Track Changes”

2. "A" to "An" on line 36. Fixed

3. "long-chain polyunsaturated fatty acids (LC-PUFA)" to "LC-PUFA" on line 72. Fixed

4. What is "temperature areas" in line 102? It is written temperate (not temperature) in line 102. Temperate area is the geographical area 40 – 60oN (and 40-60oS).

5. Please modify paragraph in lines 107-111. Paragraph is re-written to “Since salmon lice is a copepod, we hypothesized that inclusion of diatoms in the fish feed could reduce lice infestation. To test this, we offered four diets of different origins to salmon, including one with diatoms, and monitored lice infestation”. 

6. The range between 50-70000L is quite huge (lines 125-126), please make sure on this description. Oops, shall be 50 000L – 70 000L, has been changed

7. Please describe how the standard protocol of exposure was done (lines 169-170) More detailed explanation now added, i.e: “Infection of the salmon was performed using a challenge model developed with the Aquaculture Research Station in Tromsø, approved by The Norwegian Food Safety Authority (ref. id 7509). This is a bath challenge model, where the fish is challenged with a known number of lice copepodids in stagnant seawater. The research station routinely produces infective copepodids by performing controlled infections, harvesting mature eggs and incubating these until they reach the copepodid stage. The dose of infection was calculated on the basis of an expected average estimate of 10 lice per fish. The experiments were performed in the salmon louse laboratory at the research station, where factors such as water temperature are closely monitored and controlled. The water outlet from the laboratory is disinfected to prevent distribution of live copepodids to the environment. For more specific description of the model see [New ref, Skjelvareid et al., 2018]».

8. Line 177, "try" to "dry". Fixed !

9. Line 216, "International" to "international". It was Internal, not international, but fixed from Internal to internal.

10. Please use dot (.) not comma (,) for decimal point through the whole manuscript. Have checked and changed this!

11. The format of describing company (e.g. see lines 242 and 254 on page 11 for Sigma Aldrich). Have fixed!

12. Line 295, "Statment" to"Statement". Fixed!

13. The description on lines 311-312 is wrong (e.g. fish fed with diet A and D was significant higher than fish that fed with diet B and C for T1 to T2), please correct it. It has been changed to the correct statement: “Fish fed diets A and D had overall significantly higher SGR (nested ANOVA, p < 0.001) than fish fed diet B and C for T0 to T3, i.e. for whole period, days 0 – 67) (Table 2, Fig 1)….

14. Please provide the content of carbohydrate in Table 3. Carbohydrate was not measured, this since it is normally not done here defining food quality. However, if capacity/competence had been present it should possibly have been included.

15. The description on lines 366-367 is wrong, please correct it. “Oops, changed to: “Diet A (Algae) and C (Calanus) were identical with respect to amino acid content, except for glutamic acid (Glu) that was significantly lower in diet A than in the other diets”

16. Line 369, "Tyrosine" to "tyrosine". Fixed!

17. Line 405, "EEA" to "EAA". 

 Fixed!

18. Line 422, "were" to "higher" (I am not sure)? 

 Corrected to: “Fish fed diet A and B were higher in in omega-3 PUFAs (EPA + DHA) compared to the other diets. Diet C resulted in the lowest amounts of omega-3 PUFAs, i.e. less than half of the content in fish fed diets A and B”.

19. Line 453, "[42] [43,44]" to "[42-44]". Fixed!

20. Lines 455-458. The authors should not summarize higher growth in this manuscript compared to other previous studies is due to good health. Does that mean the fish in other studies are all in bad health? Ok, we “admit” this shall be toned down, and have changed the actual paragraph to: “Our fish hence performed in the high end of obtainable growth for salmon at the same weight and temperature [46]. The SGR range between the four diet groups also varied little (1.29 to 1.44 % day-1), and we interpret this as an indication of that the fish were at acceptable health compared to what is common in salmon aquaculture”.

21. Lines 488 and 490, I can't see Fig.3 and Fig. 4. Are they suppose to be Table 5 and Table 8, respectively? Correct, this has been changed!!

22. What is "though stands" (Line 544)?. This is in the Conclusion section, that now has been changed to: “Our investigation showed statistically significantly reduced lice infestation in salmon offered conventional feed with diatom biomass. In our investigation there were no correlations between feed ingredients and lice infestation. Neither were we able to detect a bioactive compound that was active as a lice deterrent, nor could we relate differences in lice infestation to health differences. The main result that lice infestation was reduced in the salmon group offered diatom biomass though still indicates the presence of some active lice deterring ingredient. In addition to that the ingredient may come from the feed, it can also have been synthesized or resulted from chemical reactions in the fish. We therefore strongly believe that further investigations should be undertaken in order to get more insight into the potential production and secretion of invertebrate repelling aldehydes in salmon, and that experiments should also take place in open sea pens under natural conditions”.

23. Line 545, "healt" to "health". Ok!

24. Please organize the format for "Reference". Done!

25. The language in supplementary data should switch to English Ok, done!

22.05.2021

Hans Chr. Eilertsen (sign.)

---

## [Decision Letter · Decision Letter 1]

22 Jun 2021

PONE-D-21-08390R1

Inclusion of photoautotrophic cultivated diatom biomass in salmon feed can deter lice

PLOS ONE

Dear Dr. Eilertsen,

Thank you for submitting your manuscript to PLOS ONE. After careful consideration, we feel that it has merit but does not fully meet PLOS ONE’s publication criteria as it currently stands. Therefore, we invite you to submit a revised version of the manuscript that addresses the points raised during the review process.

1. The data cannot support the conclusions. PLOS ONE is designed to communicate primary scientific research, and welcome submissions in any applied discipline that will contribute to the base of scientific knowledge. But the data of this manuscript cannot support the conclusions. 

We look forward to receiving your revised manuscript.

Kind regards,

Tzong-Yueh Chen, Ph.D.

Academic Editor

PLOS ONE

Reviewers' comments:

Reviewer's Responses to Questions

**Comments to the Author**

1. If the authors have adequately addressed your comments raised in a previous round of review and you feel that this manuscript is now acceptable for publication, you may indicate that here to bypass the “Comments to the Author” section, enter your conflict of interest statement in the “Confidential to Editor” section, and submit your "Accept" recommendation.

Reviewer #1: (No Response)

Reviewer #2: All comments have been addressed

2. Is the manuscript technically sound, and do the data support the conclusions?

Reviewer #1: Yes

Reviewer #2: Yes

3. Has the statistical analysis been performed appropriately and rigorously? 

Reviewer #1: Yes

Reviewer #2: Yes

4. Have the authors made all data underlying the findings in their manuscript fully available?

Reviewer #1: Yes

Reviewer #2: Yes

5. Is the manuscript presented in an intelligible fashion and written in standard English?

Reviewer #1: Yes

Reviewer #2: Yes

6. Review Comments to the Author

Reviewer #1: The authors did not respond to the two remarks "2. The potential indicator or the active compound for reducing lice in diatom were necessary to achieve the reproducibility in the present study." and "3. The authors assessed the composition in diets and salmon, and those results were further used to analyze the correlation to lice infestation. Unfortunately, there was no direct linkage between the decreased lice infestation and diatom.". These were not questions, but comments.

Reviewer #2: The revised manuscript is with better quality now, but still some issues need to be addressed.

1. In the introduction the authors stated that “oxylipin” is a group of copepod grazing deterrents in marine ecosystems. But why the authors did not detect oxylipin in the algae, feed or fish fillets?

2. The inclusion of marine oil in Norwegian fish feed has the resent years decreased to “The inclusion of marine oil in Norwegian fish feed in the recent years has decreased” in line 59.

3. In line 69, “affect” to “effect”.

4. Line 80, reference is required.

5. Line 477, Fish fed diet A and B were higher in in omega-3 PUFAs (EPA + DHA) compared to the other diets. However, in Table 8, the amount of EPA+DHA was group B (0.64) > group A (0.50) > group D (0.40) > group C (0.19). Therefor this statement is wrong.

6. Line 479, “i.e. less than half of the content in” seems an unfinished sentence.

7. PLOS authors have the option to publish the peer review history of their article (what does this mean?). If published, this will include your full peer review and any attached files.

Reviewer #1: No

Reviewer #2: No

---

## [Author Response · Author response to Decision Letter 1]

26 Jun 2021

Reviewers comments Author responses

Reviewer #1: . 

The authors did not respond to the two remarks "2. The potential indicator or the active compound for reducing lice in diatom were necessary to achieve the reproducibility in the present study." and "3. The authors assessed the composition in diets and salmon, and those results were further used to analyze the correlation to lice infestation. Unfortunately, there was no direct linkage between the decreased lice infestation and diatom.". These were not questions, but comments. We understand that these points were comments but will anyway answer/comment to these points because they points to important issues in the manuscript. Our statements here are that we, before this investigation was performed, had a hypothesis that there was some ingredient (here oxylipin) that could cause reduced lice infestation. We though were unable to detect such a compound. The question is then what caused this, and our answer here is that it obviously was not some general health issues related to poor growth or feed ingredient. What was exclusive for the salmon that experienced reduced lice was feed with algae inclusion. We therefore only can give “circumstantial” evidence that it was something in the algae that caused this directly or that it was algae ingredients that triggered production of e.g. an aldehyde in the salmon. We have tried to even clearer in the abstract and conclusion here. Now we state in abstract: “What was exclusive for salmon that experienced reduced lice was diatom inclusion in the feed. This therefore still indicates the presence of some lice deterring ingredient, either in the feed, or an ingredient can have triggered production of an deterrent in the fish”.

Reviewer #2: 

1. In the introduction the authors stated that “oxylipin” is a group of copepod grazing deterrents in marine ecosystems. But why the authors did not detect oxylipin in the algae, feed or fish fillets? See comment (to reviewer 1) above, i.e. we were not able to detect some active substance (oxylipin, group aldehyde), but it may be due to that such ingredients are unstable. The results therefore do not exclude that some deterrent was active, but the only “hard evidence” was the presence of diatoms in the feed that caused reduced lice infestation. We as mentioned above have made this clearer both in abstract and conclusion! 

 2. The inclusion of marine oil in Norwegian fish feed has the resent years decreased to “The inclusion of marine oil in Norwegian fish feed in the recent years has decreased” in line 59. This has been fixed!

3. In line 69, “affect” to “effect”. This has been fixed!

4. Line 80, reference is required. References no. 1 and 8 should be here, has been added!

5. Line 477, Fish fed diet A and B were higher in in omega-3 PUFAs (EPA + DHA) compared to the other diets. However, in Table 8, the amount of EPA+DHA was group B (0.64) > group A (0.50) > group D (0.40) > group C (0.19). Therefor this statement is wrong. Changed to: «Fish fed diet B were highest in omega-3 PUFAs (EPA + DHA), i.e. 0.64g/100g flesh, followed by diet A (0.50), D (0.40) and C (0.19)”.

6. Line 479, “i.e. less than half of the content in” seems an unfinished sentence. Changed to: « Diet C resulted in the lowest amounts of omega-3 PUFAs, i.e. less than half of the content in the other treatments”.

Else there are some other minor linguistic changes implemented 

Ethics statement appears two places! 26.06.21… removed Ethics statement from initial fish experiment description and ensured it is only in Methods under separate heading!

26.06.2021

Hans Chr. Eilertsen (sign.)

---

## [Decision Letter · Decision Letter 2]

15 Jul 2021

Inclusion of photoautotrophic cultivated diatom biomass in salmon feed can deter lice

PONE-D-21-08390R2

Dear Dr. Eilertsen,

We’re pleased to inform you that your manuscript has been judged scientifically suitable for publication and will be formally accepted for publication once it meets all outstanding technical requirements.

Kind regards,

Tzong-Yueh Chen, Ph.D.

Academic Editor

PLOS ONE

Additional Editor Comments (optional):

Reviewers' comments:

Reviewer's Responses to Questions

**Comments to the Author**

1. If the authors have adequately addressed your comments raised in a previous round of review and you feel that this manuscript is now acceptable for publication, you may indicate that here to bypass the “Comments to the Author” section, enter your conflict of interest statement in the “Confidential to Editor” section, and submit your "Accept" recommendation.

Reviewer #1: (No Response)

Reviewer #2: All comments have been addressed

2. Is the manuscript technically sound, and do the data support the conclusions?

Reviewer #1: Yes

Reviewer #2: Yes

3. Has the statistical analysis been performed appropriately and rigorously? 

Reviewer #1: Yes

Reviewer #2: Yes

4. Have the authors made all data underlying the findings in their manuscript fully available?

Reviewer #1: Yes

Reviewer #2: Yes

5. Is the manuscript presented in an intelligible fashion and written in standard English?

Reviewer #1: Yes

Reviewer #2: Yes

6. Review Comments to the Author

Reviewer #1: Although the active compounds in algae were not further estimated, yet the authors pointed out a potential candidate for controlling lice in salmon cultivation for the following studies. The MS can be considered to be accepted.

Reviewer #2: The authors have now completed all the corrections and I think it is now acceptable to be published.

7. PLOS authors have the option to publish the peer review history of their article (what does this mean?). If published, this will include your full peer review and any attached files.

Reviewer #1: No

Reviewer #2: No

---

## [Editor Report · Acceptance letter]

21 Jul 2021

PONE-D-21-08390R2 

Inclusion of photoautotrophic cultivated diatom biomass in salmon feed can deter lice 

Dear Dr. Eilertsen:

I'm pleased to inform you that your manuscript has been deemed suitable for publication in PLOS ONE. Congratulations! Your manuscript is now with our production department. 

Kind regards, 

on behalf of

Prof. Tzong-Yueh Chen 

Academic Editor

PLOS ONE